# MiR-630 Promotes Radioresistance by Induction of Anti-Apoptotic Effect via Nrf2–GPX2 Molecular Axis in Head–Neck Cancer

**DOI:** 10.3390/cells12242853

**Published:** 2023-12-17

**Authors:** Guo-Rung You, Ann-Joy Cheng, Eric Yi-Liang Shen, Kang-Hsing Fan, Yi-Fang Huang, Yu-Chen Huang, Kai-Ping Chang, Joseph T. Chang

**Affiliations:** 1Department of Medical Biotechnology and Laboratory Science, College of Medicine, Chang Gung University, Taoyuan 33302, Taiwan; d000017007@cgu.edu.tw (G.-R.Y.); annjoycheng@gap.cgu.edu.tw (A.-J.C.); 2Graduate Institute of Biomedical Sciences, College of Medicine, Chang Gung University, Taoyuan 33302, Taiwan; 3Department of Radiation Oncology and Proton Therapy Center, Linkou Chang Gung Memorial Hospital, Taoyuan 333423, Taiwan; pts@cgmh.org.tw (E.Y.-L.S.); kanghsing@adm.cgmh.org.tw (K.-H.F.); 4Department of Radiation Oncology, New Taipei Municipal TuCheng Hospital, New Taipei City 236017, Taiwan; 5Department of General Dentistry, Linkou Chang Gung Memorial Hospital, Taoyuan 333423, Taiwan; a7506@adm.cgmh.org.tw; 6Graduate Institute of Dental and Craniofacial Science, College of Medicine, Chang Gung University, Taoyuan 33302, Taiwan; 7School of Dentistry, College of Oral Medicine, Taipei Medical University, Taipei 11031, Taiwan; 8Department of Oral and Maxillofacial Surgery, Linkou Chang Gung Memorial Hospital, Taoyuan 333423, Taiwan; circlex@adm.cgmh.org.tw; 9Department of Otorhinolaryngology, LinKou Chang Gung Memorial Hospital, Taoyuan 333423, Taiwan; changkp@cgmh.org.tw; 10School of Medicine, Chang Gung University, Taoyuan 33302, Taiwan

**Keywords:** miR-630, head and neck cancer (HNC), radioresistance, apoptosis, Nrf2, GPX2

## Abstract

Head and neck cancer (HNC) ranks among the top ten prevalent cancers worldwide. Radiotherapy stands as a pivotal treatment component for HNC; however, radioresistance in cancerous cells often leads to local recurrence, becoming a substantial factor in treatment failure. MicroRNAs (miRNAs) are compact, non-coding RNAs that regulate gene expression by targeting mRNAs to inhibit protein translation. Although several studies have indicated that the dysregulation of miRNAs is intricately linked with malignant transformation, understanding this molecular family’s role in radioresistance remains limited. This study determined the role of miR-630 in regulating radiosensitivity in HNC. We discovered that miR-630 functions as an oncomiR, marked by its overexpression in HNC patients, correlating with a poorer prognosis. We further delineated the malignant function of miR-630 in HNC cells. While it had a minimal impact on cell growth, the miR-630 contributed to radioresistance in HNC cells. This result was supported by decreased cellular apoptosis and caspase enzyme activities. Moreover, miR-630 overexpression mitigated irradiation-induced DNA damage, evidenced by the reduced levels of the γ-H2AX histone protein, a marker for double-strand DNA breaks. Mechanistically, the overexpression of miR-630 decreased the cellular ROS levels and initiated Nrf2 transcriptional activity, resulting in the upregulation of the antioxidant enzyme GPX2. Thus, this study elucidates that miR-630 augments radioresistance by inducing an anti-apoptotic effect via the Nrf2–GPX2 molecular axis in HNC. The modulation of miR-630 may serve as a novel radiosensitizing target for HNC.

## 1. Introduction

Head and neck cancer (HNC) is a prevalent and challenging form of cancer worldwide. It ranks as the seventh most common type of cancer and is more prevalent in Southeast Asia. This group of cancers, including malignancies in the oral cavity, oropharynx, and paranasal sinuses, arise from specific anatomical sites. HNC is associated with several risk factors, including tobacco smoking, alcohol consumption, betel quid chewing, and human papillomavirus infection. Despite advances in treatment, the 5-year survival rate for advanced HNC remains below 50%, showing slight improvement over the past two decades [1,2].

Current treatment modalities for HNC patients include surgery, radiation, chemotherapy, or a combination of these approaches. Despite advances in treatment strategies over the last few decades, the overall 5-year survival rate for HNC patients has not significantly improved. Radiotherapy is a crucial component of curative treatment for HNC, but tumor recurrence after radiotherapy remains a significant hurdle to recovery [3]. The major cellular molecules contributing to radioresistance involve pathways related to DNA damage repair, cell survival, and apoptosis [4,5,6]. DNA repair proteins, such as Rad51 and BRCA1/2, crucial in repairing double-strand (ds) DNA damage, are frequently implicated in the development of radioresistance [4]. Additionally, signal transduction molecules within the PI3K/AKT/mTOR pathway play a critical role in maintaining cellular proliferation and viability, thereby significantly influencing the effectiveness of radiotherapy [5]. Enzymes mitigating oxidative stress, like superoxide dismutase (SOD) and glutathione peroxidase 2 (GPX2), are essential in neutralizing reactive oxygen species produced during radiation therapy, further contributing to the resilience of cancer cells against radiotherapy [6]. Conversely, targeting pro-apoptotic proteins, such as Bax and caspase 9, may increase the sensitivity of radioresistant cells, representing a promising approach to counteract radioresistance [4]. This strategy has gained importance, as many molecules associated with radiation resistance are also linked to aggressive tumor behaviors and poor patient outcomes. Identifying radioresistant molecules that contribute to a poorer prognosis may facilitate patient consultation and treatment selection, ultimately improving therapeutic outcomes.

MicroRNAs (miRNAs) are a prevalent class of small, single-stranded, non-coding RNAs (20–25 nts). Mature miRNAs have been implicated in controlling protein expression via complementary base pairing in the 3′ untranslated region of targeting mRNAs, resulting in mRNA degradation or translational inhibition [7]. Consequently, miRNAs mediate numerous cellular and physiological processes, including tissue development, cellular differentiation, proliferation, metabolic and signaling pathways, apoptosis, and stem cell maintenance [7,8]. Accumulating evidence has allowed for the screening of the expression profiles of miRNAs in HNC, and the aberrant miRNA expression levels have been associated with HNC cancer progression [9,10,11]. However, limited reports of miRNAs related to the radiosensitivity function indicate that the underlying mechanism in this context has not been solved.

MiR-630 is located at chromosome 15q24.1, near loc100420930. Currently, miR-630 has been reported in various cancers with diverse roles. It is overexpressed in multiple types of cancers, including osteosarcoma, ovarian, colorectal, bladder, hepatic, gastric, and renal cancers [12,13,14,15,16,17,18,19]. In these contexts, miR-630 functions as an oncomiR by promoting cell proliferation, migration, invasion, and increasing chemoresistance or radioresistance [20,21,22,23,24,25,26]. Conversely, miR-630 has also been reported as a tumor suppressor, as shown by the down-regulation in tumors and inhibiting cancer aggressiveness in several types of cancer, such as esophageal, breast, glioma, and lung [27,28,29,30]. Hence, miR-630 plays a pivotal role in maintaining cellular homeostasis, and its dysregulation is closely associated with various malignancies. Despite these findings, the biological function of miR-630 in HNC is still uncertain, especially regarding its radioresistant role.

In this study, we investigated the potential function of miR-630 in HNC. We discovered that miR-630 promotes radioresistance by the induction of intrinsic anti-apoptotic effects. We further characterized the molecular mechanism and found that this effect was achieved by inhibiting DNA damage response via the Nrf2–GPX2 molecular pathway. The potential application of miR-630 in prognosis or radiosensitization for HNC was revealed.

## 2. Materials and Methods

### 2.1. Patients and Plasma miR-630 Detection

This study received approval from the Institutional Review Board of Chang Gung Memorial Hospital, Taiwan. A total of 32 non-cancer or healthy individuals and 72 HNC patients were recruited for the study. None of the HNC patients had undergone either radiotherapy or chemotherapy before surgery. The clinical characteristics of the recruited patients are summarized in Appendix A. All the participants provided written informed consent, indicating their willingness to donate plasma for clinical research.

For each recruited individual or patient, EDTA-plasma samples were collected. Plasma miRNAs were then purified from 200 μL of the plasma specimens using an miRNeasy Mini Kit (Qiagen, Germantown, MD, USA) similarly as previously described [31]. The purified miRNAs were enriched in 20 μL of nuclease-free water. MiRNA determinations were carried out using specific stem-loop RT primers from a TaqMan^®^ miRNA assays kit (Applied Biosystems, Waltham, MA, USA). The levels of miRNA expression were detected by RT-qPCR, following the manufacturer’s protocol. The PCR reactions were carried out using a Bio-Rad CFX96 detection system (Bio-Rad, Hercules, CA, USA). The relative expression levels, presented as fold changes, were determined by comparing them to the levels of endogenous miR-630 in OECM1 cancer cells.

The KM-Plotter online tool (http://kmplot.com/analysis, accessed on 10 October 2023) was used to assess the prognostic significance of miR-630 in the TCGA-HNSC cohort (*n* = 522). The patients were classified into high- and low-risk groups using an optimization algorithm, and their overall survival was evaluated using a Kaplan–Meier analysis. Hazard ratios with 95% confidence intervals were calculated to quantify the differences in survival between these groups.

### 2.2. Cell Line and Cell Culture

HNC lines OECM1, SAS, and FaDu were used in this study. The specific information of each cell line, including the accession number of the database and the cell culture condition, has been previously described [32]. Briefly, the OECM1 cells were grown in RPMI 1640 medium (Thermo Fisher Scientific, Waltham, MA, USA), while the FaDu and SAS cells were cultured in MEM and DMEM medium (Thermo Fisher Scientific, Waltham, MA, USA), respectively. The culture media were supplemented with 10% fetal bovine serum (FBS) and 1% antibiotic antimycotic. All the cell lines were maintained at 37 °C in a humidified atmosphere containing 5% CO_2_ air.

### 2.3. MiR-630 Mimic Oligonucleotides and Transfection

The miR-630 mimic (sense: 5′-AGUAUUCUGUACCAGGGAAGGU-3′; antisense: 5′-CUUCCCUGGUACAGAAUACUUU-3′) and its corresponding control oligonucleotides (sense: 5′-UUCUCCGAACGUGUCACGUTT-3′; antisense: 5′-ACGUGACACGUUCGGAGAATT-3′) were designed and synthesized by GenePharma (Pudong New Area, Shanghai, China). For cell transfection, the cells were transfected with the miRNA mimic or the control using jetPRIME^®^ transfection reagent (Polyplus-transfection, Illkirch, France), following the manufacturer’s protocol. The cellular functions were assessed 24 h post-transfection.

### 2.4. Determination of miRNA or mRNA Expressions by RT-qPCR Method

The process of RNA extraction and mRNA determination followed procedures similar to the previous study [33]. In summary, the total RNA was extracted from cells using TRizol reagent (Invitrogen, Carlsbad, CA, USA). Subsequently, cDNA synthesis and PCR reactions were carried out using SyBr Green Supermix reagent (Bio-Rad, Hercules, CA, USA) and detected by a Bio-Rad CFX96 system. For the analysis of miRNAs, the miRNAs were enriched with the cells using TRizol reagent. We determined the levels of miRNA expression by RT-qPCR using a TaqMan^®^ miRNA assay kit (Applied Biosystems, Waltham, MA, USA), following the manufacturer’s protocol. The PCR reactions were carried out using a Bio-Rad CFX96 detection system, with U6 expression serving as an internal control. The primers used in this study are listed in Appendix A.

### 2.5. Determination of Protein Expressions by Western Blot Analysis

The process of cellular protein extraction and Western blot assays was carried out following previously established protocols as described [34]. Briefly, the cells were homogenized in CHAPS lysis buffer and allowed to incubate on ice for a duration of 30 min. Subsequently, cellular proteins were obtained through centrifugation. For the Western blot analysis, 30 μg of the total cellular protein extract was loaded onto an SDS-polyacrylamide gel for electrophoresis. Following gel electrophoresis, the proteins were transferred to a protein nitrocellulose hybridization transfer membrane, which was then incubated with a specific primary antibody, followed by a secondary antibody conjugated with horseradish peroxidase. The membrane was treated with an ECL developing solution (Merck Millipore, Burlington, MA, USA) and exposed to X-ray film. The β-actin expression served as an internal control. The antibody information used in the experiments is listed in Appendix A.

### 2.6. Assessment of Cell Proliferation and Radiosensitivity

The radiosensitivity was determined by a clonogenic cell survival assay, similar to that previously described [35]. In this assay, the test cells were initially seeded in 3.5 cm cell culture dishes and allowed to grow for 24 h. Subsequently, the test cells were exposed to varying doses of radiation ranging from 0 to 6 Gy and were then continuously cultured for a period of 7 to 14 days. After this incubation, the cells were stained with crystal violet, and the survival fractions were determined by calculating the ratio of the number of colonies formed to the number of cells initially seeded, multiplied by the plating efficiency. The cell proliferation was assessed by trypan blue exclusion assay. Briefly, the cells were seeded in 6-well culture plates and continuously cultured for 3 days. Each day, the cell suspensions were stained with 0.4% trypan blue (Sigma-Aldrich, Burlington, MA, USA), and the number of live cells in the population was counted using a hemocytometer.

### 2.7. Determination of Apoptotic Cells by Annexin V-Labeled Flow Cytometry

An Annexin V-FITC apoptosis detection kit (Dojindo, Kamimashiki-gun, Kumamoto, Japan) was employed to determine the levels of cell apoptosis. This method provided a simple, fluorescent-based method for distinguishing between healthy and apoptotic cells by detecting changes in the cellular morphology, as well as the alternation of cell membranes. Briefly, the cells were stained using an Annexin V-FITC apoptosis detection kit, which includes dual fluorescent dyes for the identification of early apoptotic cells using FITC-labeled Annexin V (green fluorescence) and late apoptotic/necrotic cells using propidium iodide (red fluorescence), following the manufacturer’s instructions. The nuclei were counterstained with Hoechst 33342 (Invitrogen, Carlsbad, CA, USA). The fluorescent signals were monitored using a NucleoCounter^®^ NC-3000™ image cytometer (ChemoMetec A/S, Allerod, Denmark).

### 2.8. Determination of Double-Strand DNA Breaking Status by Fluorescent Microscopy

The DNA double-strand break marker was determined using the phospho-γH2AX antibody (ab2893, Abcam, Cambridge, UK). This assay relies on the phosphorylation of histone H2AX in response to DNA-damaging agents, which recruits DNA repair signaling factors to the DNA damage foci. These foci were examined using fluorescent microscopy (LSM 510 Meta, Zeiss, Jena, Germany). Briefly, the cells were grown on an 8-well chamber slide, fixed with 4% paraformaldehyde, permeabilized with 0.5% Triton X-100, and blocked with 1% FBS for 30 min. Subsequently, the cells were stained with a specific antibody against phospho-γH2AX overnight at 4 °C. After the overnight incubation, the cells were washed with PBS and then stained with a secondary antibody, anti-rabbit IgG conjugated with Alexa Fluor^®^ 488 (A21206, Invitrogen, CA, USA), for 1 h at room temperature. The chamber slides were then mounted with a medium containing DAPI (Vector Laboratories, Newark, CA, USA). Finally, immunofluorescence images were visualized using an inverted confocal microscope (LSM 510 Meta, Zeiss, Jena, Germany), and the mean fluorescence intensity was calculated using Image J software v1.53.

### 2.9. Determination of Intrinsic Apoptosis Associated with Mitochondrial Permeabilization

A lipophilic dye, JC-1 (ChemoMetec A/S, Allerod, Denmark), was employed to determine the permeabilization of mitochondrial potential. This method provides a fluorescent-based approach for distinguishing between healthy and apoptotic cells by detecting changes in the mitochondrial transmembrane potential. Briefly, the test cells were incubated in JC-1 solution, which contains fluorescent dyes differentially captured by healthy (red, aggregation of mitochondria) or apoptotic (green, monomer of mitochondria) cells. All the cells were stained with the Hoechst 33342 reagent for cell counting, and the fluorescent signals were analyzed using the NucleoCounter^®^ NC-3000™ image cytometer (ChemoMetec A/S, Allerod, Denmark).

### 2.10. Determination of Cellular Reactive Oxygen Species (ROS) Level

The cellular ROS level was measured using an indicator, H_2_DCFDA (Invitrogen, Carlsbad, CA, USA), following a similar protocol as previously described [36]. Briefly, the test cells were suspended in a culture medium supplemented with 10 μM H_2_DCFDA reagent for 30 min at 37 °C. During this incubation, intracellular esterases and oxidation cleaved the acetate groups, and the nonfluorescent H_2_DCFDA was converted to the highly fluorescent 2′,7′-dichlorofluorescein (DCF). After recovery of the culture medium and washing to remove the residual dye reagent, the cells were then resuspended. The fluorescence intensity of the ROS level was monitored and analyzed by guava easyCyte™ flow cytometry systems (Merck Millipore, Burlington, MA, USA).

### 2.11. Luciferase Report Assay for the Transcriptional Activity of Nrf2

A pGL3 promoter-8xARE luciferase reporter construct, which contains eight copies of antioxidant response elements (ARE) with the sequence 5′-GTGACAAAGCA-3′ in its promoter region, was used in this study, following a similar protocol as previously described [37]. It has been previously reported that the induction of ARE-driven luciferase activity is regulated by Nrf2 [37]. The plasmid construct was designed as described and purchased from MDbio Inc. (Taipei, Taiwan). Briefly, the test cells were co-transfected with the pGL3 promoter-8xARE plasmid and either the miR-630 mimic or control. After 24 h of transfection, the cells were lysed. The luciferase activity was measured in the lysed cells using the Luciferase Assay System and GloMax^®^ 20/20 Luminometer (Promega, Madison, WI, USA), following the manufacturer’s instructions. The relative luciferase activity, presented as a fold change, was determined by comparing it to the luciferase activity level in the control cells.

## 3. Results

### 3.1. MiR-630 Is Overexpressed in Patients with HNC and Associated with Poor Prognosis

To determine the relevance of miR-630 expression in HNC, we compared the miR-630 levels in the plasma samples from patients with HNC and healthy individuals. As shown in Figure 1A, the miR-630 levels were significantly higher in the HNC patients, with an approximately 2.4-fold increase compared to the healthy individuals (*p* < 0.001). To assess the differential power of miR-630, we conducted a receiver operating characteristic (ROC) analysis. As shown in Figure 1B, miR-630 had an area under the curve (AUC) of 0.7063, suggesting this molecule possesses an effective power to distinguish HNC patients from healthy individuals. To extend the effect of miR-630 on prognosis, we examined the association between the miR-630 expression levels and patient survival using the TCGA-HNSC cohort (*n* = 522). As shown in Figure 1C, high levels of miR-630 were significantly correlated with poor survival (*p* = 0.0016, HR = 1.55). These results suggest that miR-630 may be a crucial oncomiR in HNC to regulate cancer progression.

### 3.2. MiR-630 Has a Minimal Effect on Cell Growth but Promotes Radioresistance in HNC Cells

To investigate the role of miR-630 in HNC, we conducted experiments involving the transfection of HNC cells with miR-630 mimics to induce miR-630 overexpression. The effect on cell proliferation was examined using a trypan blue exclusion assay in three HNC cell lines (OECM1, SAS, and FaDu) to obtain a more general result. As shown in Figure 2A, no significant alterations in cell growth rate were observed in any of the tested cell lines, indicating that miR-630 has a minimal effect on cell growth.

We further conducted clonogenic survival assays to examine the potential effect of miR-630 on radiosensitivity. Various irradiation (IR) doses (0 to 6 Gy) were treated, and the differential survival colonies were determined. As shown in Figure 2B, the administration of miR-630 significantly increased the number of surviving colonies compared to the control groups. At a radiation dose of 6 Gy, substantial increases of 1.5-, 4.8-, and 3.6-fold were observed, respectively, in the OECM1, SAS, and FaDu cell lines. These results demonstrate that miR-630 promotes radioresistance in HNC.

### 3.3. MiR-630 Suppresses Cellular Intrinsic Apoptotic Pathways

Given that radiation-induced cellular death typically occurs through the apoptotic pathway [38], we examined whether miR-630 contributes to radioresistance via regulating this mechanism. The Annexin V staining method was used to assess the effect of miR-630 in response to IR-induced cell death. Figure 3 shows the three examined HNC cell lines; similar results were obtained. Without irradiation, miR-630 overexpression had a minimal impact on cellular apoptosis. Irradiation significantly induced cell death in the controls and the miR-630 overexpressing cells. However, the cells overexpressing miR-630 significantly reduced IR-induced cellular apoptosis, as shown by the decrease in Annexin V-positive cells to 79.3%, 80.4%, and 71.2% in the OECM1, SAS, and FaDu cells, respectively (Figure 3A). These results suggest that miR-630 contributes to radioresistance by suppressing apoptotic pathways.

While extrinsic apoptosis usually initiates following the activation of death receptors at the plasma membrane, the intrinsic pathway is often triggered by a wide range of cellular stress signals [39]. We further substantiated miR-630 mediating intrinsic apoptotic pathways by determination of the pro-apoptotic proteins Bak and Bax via Western blot analysis. As shown in Figure 3B, overexpression of miR-630 substantially inhibited both of these pro-apoptotic proteins. While it remarkably suppressed Bak by 99% in the OECM1 cells, miR-630 inhibited both Bak and Bax by 20% and 61% in the FaDu cells (Figure 3B). These results indicate that miR-630 increased cellular survival by inhibiting various pro-apoptotic signals dependent on the specific cell type. 

Caspase enzymes play the ultimate role during the apoptotic process by cleaving inactive precursors to become activated enzymes [39]. In this context, the intrinsic apoptotic associated caspases, caspase-3, -7, and -9, were examined post-IR treatment. The results are shown in Figure 3C. While there were fewer effects on the expressions of inactivated precursors, miR-630 attenuated these active enzymes. Compared to the controls, miR-630 decreased the caspase-3, -7, and -9 cleavage forms to approximately 40–60% in the OECM1 cells and 47–79% in the FaDu cells. These results demonstrate that miR-630 promotes radioresistance in HNC cells by inhibiting intrinsic apoptotic pathways via attenuating pro-apoptotic proteins to restrain caspase activity, leading to cellular survival.

### 3.4. MiR-630 Mitigates IR-Induced DNA Damage in HNC Cells

In response to radiation in cells, DNA may be damaged, leading to apoptosis. IR-induced DNA damage involves a double-stranded break, evidenced by the hallmark of overexpression of the histone variant protein γ-H2AX [40]. Thus, we conducted additional investigations to determine whether miR-630 may play a regulatory role in IR-induced DNA damage in HNC cells. The immunofluorescence staining for γ-H2AX foci in HNC cells was examined, as shown in Figure 4A. In response to IR, nearly all the cells exhibited γ-H2AX foci compared to the non-irradiated cells. However, the miR-630 overexpressed cells displayed a noticeable decrease in γ-H2AX staining compared to the control cells. The mean fluorescence intensity of residual γ-H2AX in the miR-630 overexpressed cells was significantly lower than in the control cells (*p* < 0.05). This reduction in γ-H2AX staining in the HNC cells was confirmed by the protein expression as analyzed by the Western blot method (Figure 4B). In response to IR, a substantial reduction in γ-H2AX protein levels was observed in the miR-630 overexpressed cells by 43% and 53% in two tested HNC cells. These data suggest that miR-630 promotes radioresistance by reducing the DNA damage response in HNC cells.

### 3.5. MiR-630 Attenuates Mitochondrial-Mediated ROS Level via Nrf2–GPX2 Molecular Axis

It is well established that radiation can trigger the production of reactive oxygen species (ROS) within cells, following an increase in mitochondrial permeabilization, leading to oxidative stress, DNA damage, and intrinsic apoptosis [38,39]. To address whether miR-630 may regulate mitochondrial-mediated ROS, we determined mitochondrial membrane permeabilization using the JC-1 mitochondrial membrane fluorescent staining method in the HNC cells. The results are shown in Figure 5A. After IR treatment, an increased proportion of mitochondrial depolarization cells was observed (21%) compared to the untreated cells (9%). However, this IR-induced mitochondrial depolarization effect was diminished in the miR-630-overexpressed cells (14%). The intracellular ROS level was assessed by the H_2_DCFDA staining method. Consistently, miR-630 overexpression reduced IR-induced ROS levels by 23% and 47% in two tested HNC cells (Figure 5B). These results indicate that miR-630 contributes to radioresistance by inhibiting ROS generation and mitochondrial permeability.

To explore the molecular mechanisms potentially regulated by miR-630 that lead to ROS modulation, we examined the target genes of miR-630. By employing three miRNA target prediction algorithms (TargetScan, DIANA-microT, and RNA22), we identified 255 common genes across these platforms (Appendix A). Subsequently, we performed a KEGG enrichment analysis on these genes to elucidate the potential molecular regulatory pathways. This analysis highlighted several pathways, notably, the PI3K–Akt signaling pathway, due to its significant association with carcinogenic functions [41] and its statistical relevance (*p* = 0.021) (Appendix A). The PI3K–Akt pathway plays a pivotal role in inducing Nrf2 transcriptional activity, which is essential for maintaining cellular redox balance [42]. In delving deeper into the role of miR-630 within this pathway, we found that the PTEN tumor suppressor emerged as a significant target (Appendix A). PTEN is renowned not only as a tumor suppressor, but also for its inhibitory effect on PI3K–Akt signaling [41], further positioning it as a downstream target of miR-630 [43]. Consequently, we hypothesize that miR-630 modulates ROS through the PTEN/PI3K–Akt/Nrf2 molecular axis (Appendix A).

Nrf2 is an emerging signaling molecule of a transcriptional factor responsible for transactivating genes related to antioxidant activity [44]. To further investigate the molecular effect of miR-630 in ROS modulation, we used a luciferase reporter assay with an antioxidant response element (ARE) to assess the well-known Nrf2-inducing transcriptional activity of antioxidant enzymes [37]. As shown in Figure 5C, miR-630 overexpression significantly up-regulated the ARE–luciferase activity in the two HNC cell lines. The Nrf2 transcriptional activity increased approximately 1.4-fold in the OECM1 and FaDu cells compared to the controls. These results were confirmed by the expression analysis, which determined that the Nrf2 protein was elevated in the miR-630 transfectants of the two tested HNC cell lines (Figure 5D).

To extend the Nrf2 transactivation effect, we further examined the expression of Nrf2-mediated antioxidant enzymes in two HNC cells, including the glutamate-cysteine ligase catalytic subunit (GCLC), glutamate-cysteine ligase modifier subunit (GCLM), manganese superoxide dismutase (MnSOD), and glutathione peroxidase 2 (GPX2), by RT-qPCR method. As shown in Figure 5E, while GCLC, GCLM, and MnSOD have fewer effects, the GPX2 was substantially induced following miR-630 overexpression. To confirm this modulatory effect, we examined the protein level of GPX2 by Western blot analysis. As shown in Figure 5F, this ROS scavenger enzyme was consistently increased by approximately 2-fold in two tested cells overexpressing miR-630. These results demonstrate that miR-630 attenuated cellular ROS levels by the induction of Nrf2 transactivation to up-regulate the expression of the antioxidant enzymes GPX2.

The study can be summarized through a molecular model that illustrates the miR-630 regulatory mechanism, which induces radioresistance in HNC cells (Figure 6). Overexpression of miR-630 triggers the activation of the Nrf2 molecule, subsequently promoting the upregulation of the antioxidant enzyme GPX2. This results in a reduction in cellular ROS levels, attenuating mitochondrial depolarization and diminishing IR-induced DNA damage. These, in turn, decrease the cellular intrinsic apoptotic response by inhibiting pro-apoptotic proteins (Bax and Bak) and reducing the enzyme activity of caspase 3/7/9. Collectively, these molecular alterations bestow radioresistance, which correlates with an unfavorable prognosis in HNC.

## 4. Discussion

Radiotherapy is an indispensable treatment modality in HNC, while radioresistance is the primary cause of treatment failure. MiRNA has been defined as a merging molecular family participating in malignancy, while the knowledge of miRNA in radioresistance remains limited. In this study, we unveiled the role of miR-630 in HNC, and a few points were highlighted, as follows. (1) MiR-630 is overexpressed in patients with HNC and associated with poor prognosis. (2) MiR-630 has a minimal effect on cell growth but promotes radioresistance in HNC cells. (3) MiR-630 suppresses cellular intrinsic apoptotic pathways and mitigates IR-induced DNA damage in HNC cells. (4) MiR-630 attenuates the mitochondrial-mediated ROS level via the Nrf2–GPX2 molecular axis. Future experiments should involve in vivo models to corroborate the efficacy of miR-630 as a novel radiosensitizing target for HNC.

Although miR-630 has been reported with diverse functions in various cancer types, we first addressed its role in HNC. We found that the miR-630 exhibited a significantly higher level in the plasma from HNC patients than the healthy controls (Figure 1A). This result agrees with the report that miR-630 is up-regulated in patients with nasopharyngeal cancer plasma to serve as a detection marker [45]. Recent evidence suggests that miRNAs, including miR-630, are released into the circulation, acting as a new class of cancer biomarkers in a process akin to liquid biopsy [10,45]. These cell-free miRNAs are stable in the bloodstream, either encapsulated in extracellular microvesicles or bound to the Arg2 protein, thus evading degradation by endogenous ribonucleases [11]. Our findings reveal that the plasma levels of miR-630 are significantly elevated in HNC patients compared to healthy controls. These findings also highlight the potential of miR-630 as a promising biomarker for HNC. We further note that a higher level of miR-630 correlates with worse survival in HNC (Figure 1C). This result is consistent with the findings in renal, gastric, ovarian and bladder urothelial cancers that up-regulation of this molecule is associated with poor overall survival [16,19,46,47]. Functionally, miR-630 promotes radioresistance in HNC cells (Figure 2), supported by similar investigations in cervical [25,26]. Thus, miR-630 plays an oncogenic role in HNC, which promotes radioresistance and predicts a worse prognosis.

Tumor resistance to IR requires cells to undergo apoptotic regulation, DNA repair, and maintenance of their oxidative status, which collectively contribute to the acquisition of radioresistant features [48]. The activation of cell apoptosis is the primary response to cell death following radiotherapy treatment. Cell apoptosis involves two main pathways: the intrinsic mitochondrial pathway and the extrinsic death receptor pathway [49]. Extrinsic apoptosis is initiated following the activation of death receptors at the plasma membrane by their cognate extracellular death ligands, while the intrinsic pathway is triggered by a wide range of cellular stress signals, resulting in the induction of mitochondrial outer membrane permeabilization and a series of caspase activations, including caspase 9 and caspase 3 [39]. The BCL-2 protein family also modulates intrinsic apoptosis that comprises pro- and anti-apoptotic members. The pro-apoptotic members, including Bax and Bak, stimulate this mitochondrial function, whereas anti-apoptotic members, such as Bcl-2, antagonize this activity [50]. In this study, we showed that miR-630 plays a critical role in suppressing the intrinsic apoptotic pathway, as evidenced by the reduced IR-induced cell death (Figure 3A), decreased pro-apoptotic proteins Bak and Bax (Figure 3B), and reduced caspase activity involved in intrinsic pathways, caspase 3/7/9 (Figure 3C). These results are supported by other investigations found in colorectal, renal, and ovarian cancers [15,43,51]. Thus, miR-630 inhibits cellular apoptosis, which may possess potential as an alternative strategy for cancer treatment.

DNA damage has been demonstrated to have significant implications in cancer, especially in response to radiotherapy. Major types of DNA damage include intra/inter-strand crosslinks, single-strand breaks, and double-strand breaks (DSBs), with DSBs being the primary target of radiation-induced DNA damage [52]. Histone H2AX has been implicated as a marker of DSBs, as its reduction is associated with radioresistance [40,53]. In the present study, we found that miR-630 overexpression can reduce γH2AX protein levels in both whole cell lysates and the cell nucleus (Figure 4). This result is supported by previous findings that miR-630 inhibited the DNA damage response, including downregulation of RAD18 or MCM8 repairing proteins or blocking the phosphorylation of the ataxia-telangiectasia mutated (ATM) kinase and two ATM substrates, histone H2AX and p53 [54,55]. Thus, miR-630 may be a novel factor in regulating the DNA damage response. Moreover, severe DNA damage can trigger ROS production, leading to the activation of p53, which subsequently regulates cellular apoptosis by modulating Bax and caspase activity [56]. Hence, miR-630 might emerge as a novel factor in the regulation of the DNA damage response by reducing ROS production, consequently leading to diminished cellular apoptosis.

In physiological conditions, redox homeostasis is achieved by constantly balancing cellular reactive oxidative stress (ROS) and the scavenging mechanisms [57]. Mitochondria-dependent ROS production is the most crucial factor leading to cell death after IR, while activation of ROS scavenger pathways is tightly associated with radioresistance [36,58]. ROS is induced by a burst when cells endure severe DNA damage, such as that caused by exposure to ionizing radiation, like X-rays or high-power microwaves, which may lead to cellular apoptosis [56]. Therefore, we investigated whether miR-630 contributes to radioresistance via induction of the ROS regulatory mechanism. The present study shows that miR-630 can reduce mitochondrial permeability and cellular ROS levels in HNC cells in response to radiation (Figure 5A,B). This reduction in cellular ROS may be the up-stream mechanism of weakened DNA damage and diminished intrinsic apoptosis following radiation treatment.

Nrf2 is a critical molecular regulator of antioxidant protection to correct cellular ROS imbalance, including glutathione production, utilization, and regeneration [59]. Biochemically, Nrf2 functions as a transcriptional factor that activates the expressions of several ROS scavenger enzymes, such as NQO1, GCLC, GCLM, and GPX2. [60]. Recently, the inhibition of the Keap1–Nrf2 interaction and the activation of Nrf2 interacting with NF-κB, PI3K/Akt, Notch, MAPK, and Wnt signaling pathways have been investigated [42]. The oncogenic PI3K/Akt signaling pathway, a crucial survival pathway activated in cancer [41], is negatively regulated by the loss of the tumor suppressor PTEN. This loss is one of the major factors resulting in cancer progression, treatment resistance, and poor outcomes [61,62,63]. In this study, integrating miRNA target prediction data with a pathway enrichment analysis suggests that miR-630 may regulate PTEN expression in the PI3K/Akt pathway (Appendix A). This hypothesis is supported by previous results indicating that miR-630 inhibits apoptosis and contributes chemoresistance in ovarian carcinoma by targeting PTEN [43]. Intriguingly, the loss of PTEN also leads to increased Nrf2 protein expression and activity, resulting in carcinogenesis [64]. Therefore, PTEN might serve as a pivotal regulator for Nrf2. In this study, we also demonstrated that miR-630 promotes Nrf2 transcriptional activity and up-regulation of GPX2 expression in HNC cells (Figure 5). This enzyme activation may increase ROS clearance, reduce ROS levels, and lead to stress tolerance. Our findings are consistent with other investigators. The activation of Nrf2 and its downstream enzyme GPX2 may promote radioresistance or cisplatin resistance in HNC or lung cancer [37,65,66,67]. Furthermore, miR-630 contributes to radio- or drug- resistance in various cancers [15,22,23,26,43]. 

MicroRNAs (miRNAs) have recently become promising therapeutic targets in cancer treatment [68,69,70]. The development of therapeutics that either inhibit oncogenic miRNAs or enhance the expression of tumor-suppressor miRNAs holds substantial promise [69]. Notably, advancements in antisense oligonucleotides, especially chemically modified types, like locked nucleic acid oligonucleotides, have shown more effectiveness in silencing aberrant miRNAs [71]. Additionally, RNA-targeted therapeutic approaches, including small interfering RNA (siRNA) technologies, are advancing into clinical trials [68,69,72]. These techniques have demonstrated high efficacy and potentially fewer side effects than conventional chemoradiation therapies [68,69]. In the present study, we demonstrated that miR-630 induces the transcriptional activation of Nrf2, promoting GPX2 expression to attenuate mitochondria damage and ROS levels, resulting in resistance to IR-induced DNA damage and cellular apoptosis. Since miR-630 is overexpressed in HNC and contributes to radioresistance, targeting this miRNA using siRNA-based methods could be a potential strategy. By silencing miR-630, it may be possible to amplify the ROS-inducing apoptotic effects and enhance the sensitivity of cancer cells to radiotherapy. Such an approach may improve the overall efficacy of radiotherapeutic treatments in HNC, offering more targeted and effective management of the disease.

## 5. Conclusions

This study elucidates the specific role of miR-630 in promoting radioresistance in HNC. We delved deeper to characterize the underlying mechanism, revealing the involvement of multiple cellular processes. These included the suppression of intrinsic apoptotic pathways, reduction in the DNA damage response, and the inhibition of mitochondrial-mediated ROS levels via the activation of the Nrf2–GPX2 pathway. Our findings pave the way for improved radiotherapy strategies, highlighting the potential of targeting miR-630 and its related pathway in the treatment of refractory HNC.

## Figures and Tables

**Figure 1 cells-12-02853-f001:**
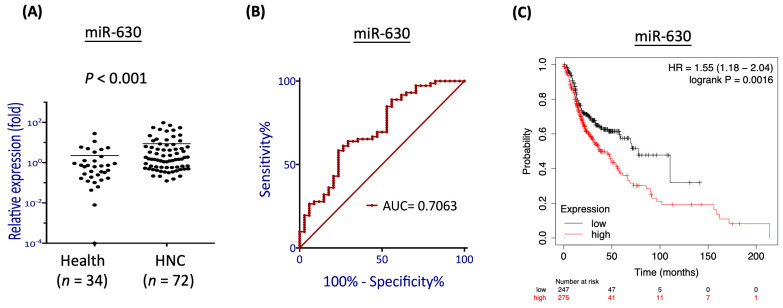
MiR-630 is overexpressed in patients with head and neck cancer (HNC) and associated with poor prognosis. A total of 104 plasma samples from 32 normal individuals and 72 HNC patients were recruited. (**A**) The scatter dot plot shows the relative expression levels of miR-630 in healthy individuals and HNC patients. (**B**) ROC analyses were performed to assess the level of miR-630 across samples from healthy individuals and HNC patients with cancer. The AUC was used to determine the degree of separation between these groups. (**C**) Prognostic significance of miR-630 in HNC patients, as determined by the Kaplan–Meier Plotter online tool using the TCGA-HNSC dataset (*n* = 522).

**Figure 2 cells-12-02853-f002:**
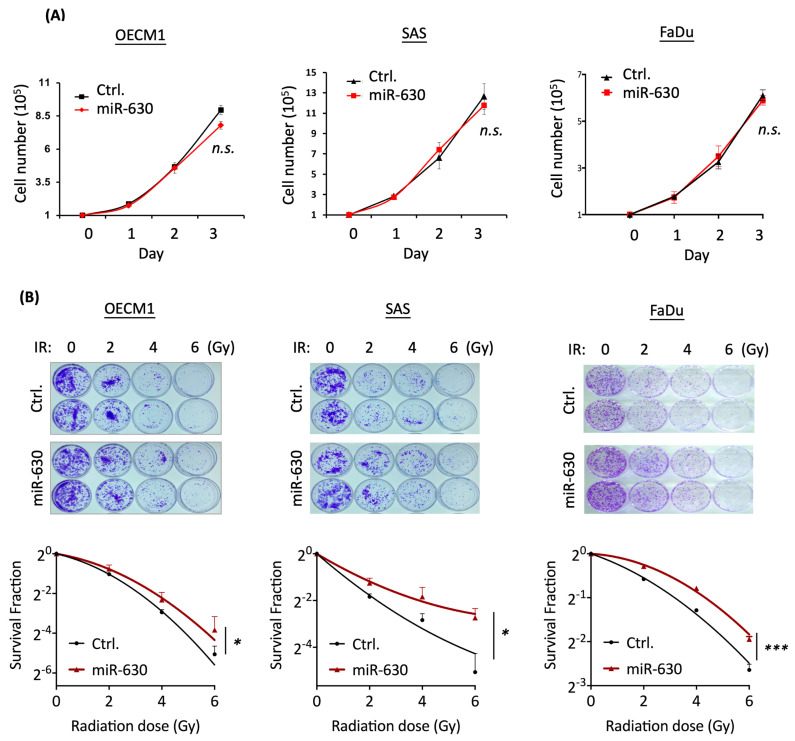
MiR-630 has a minimal effect on cell growth but promotes radioresistance in HNC cells. Overexpression of miR-630 led to slight regulation of cell growth but promoted radioresistance in HNC cells. (**A**) Following transfection with the miR-630 mimic, HNC cells (OECM1, SAS, and FaDu) were subjected to cell growth assays. (**B**) MiR-630 increased radioresistance in HNC cells. HNC cells (OECM1, SAS, and FaDu) were transfected with the miR-630 mimic (miR-630) or control (Ctrl.) 24 h before IR treatment, and the cells were then subjected to clonogenic cell survival assays. The colony survival fractions were determined after IR treatment with various doses (0 to 6 Gy). All data are presented as mean ± SD of three independent experiments (n.s., no significant; *, *p* < 0.05; ***, *p* < 0.001; ANOVA test).

**Figure 3 cells-12-02853-f003:**
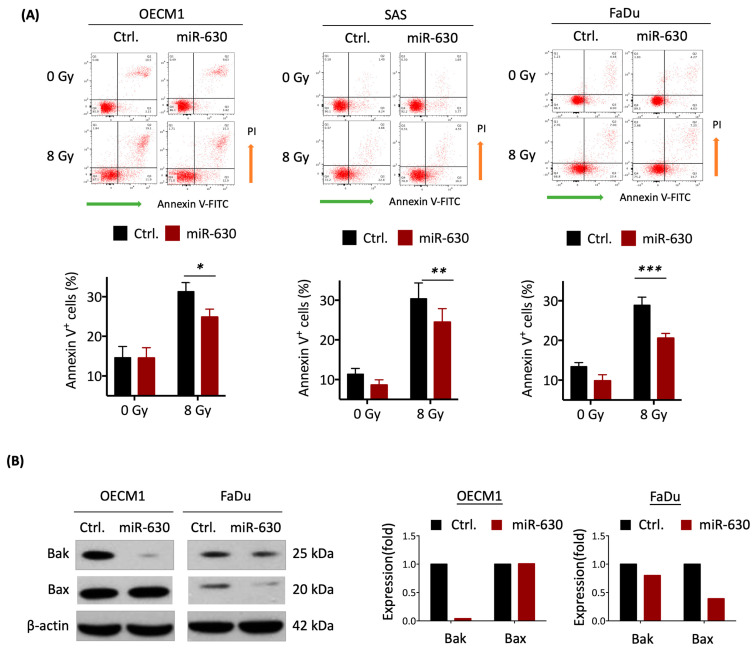
MiR-630 suppresses cellular intrinsic apoptotic pathways. MiR-630 decreases cell apoptosis following IR exposure. (**A**) HNC cells (OECM1, SAS, and FaDu) were pre-transfected with the miR-630 mimic (miR-630) or control (Ctrl.) 24 h before 0 or 8 Gy IR treatment. Cell apoptosis was measured using Annexin V staining in HNC cells at 48 h after IR treatment. Data are presented as mean ± SD of three independent experiments (*, *p* < 0.05; **, *p* < 0.01; ***, *p* < 0.001; *t*-test). (**B**,**C**) Pro-apoptosis proteins (Bax and Bak) and apoptosis-related caspases (precursor-form (P) and cleaved-form (**C**) of caspase-3/-7/-9) were assessed using Western blot to detect the levels of cleavage enzymes at 72 h after 8 Gy IR treatment. β-actin was used as a loading control. HNC cells (OECM1 and FaDu) were pre-transfected with the miR-630 mimic or control 24 h before IR treatment. (**B**) MiR-630 inhibits pro-apoptosis proteins. (**C**) MiR-630 suppressed the enzyme activities of caspase-3/-7/-9.

**Figure 4 cells-12-02853-f004:**
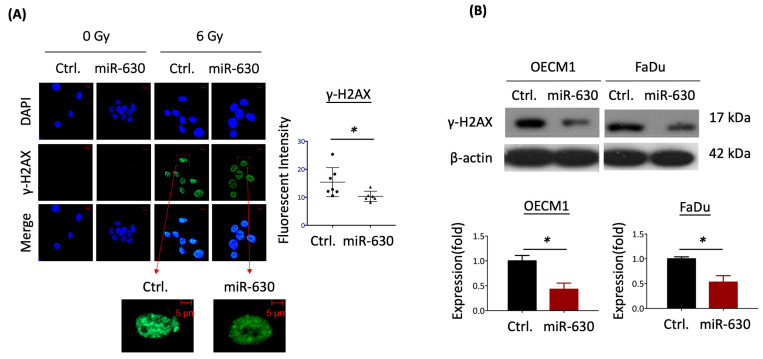
MiR-630 mitigates IR-induced DNA damage in HNC cells. (**A**) MiR-630 decreased radiation-induced DNA-break marker γ-H2AX. Photographs of γ-H2AX immunofluorescence staining (green) in FaDu cells transfected with miR-630 mimic (miR-630) or control (Ctrl.) before 0 or 6 Gy IR treatment. (Scale bar = 20 µm). Zoom of γ-H2AX expression was visualized after confocal microscopy. (Scale bar = 5 µm). Right: The quantitative results of mean fluorescence intensity were calculated using Image J. Data are presented as mean ± SD of three independent experiments (*, *p* < 0.05; *t*-test). (**B**) MiR-630-attenuated IR-induced DNA damage response. The HNC cells (OECM1 and FaDu) were pre-transfected with miR-630 mimic or control 24 h before IR treatment. The protein level of γ-H2AX was assessed using Western blot to detect the levels of DNA damage response at 24 h after 6 Gy IR. The β-actin was used as a loading control. The lower panel shows the quantitative results of the protein bands determined by Image J. Data are presented as mean ± SD of three independent experiments (*, *p* < 0.05, *t*-test).

**Figure 5 cells-12-02853-f005:**
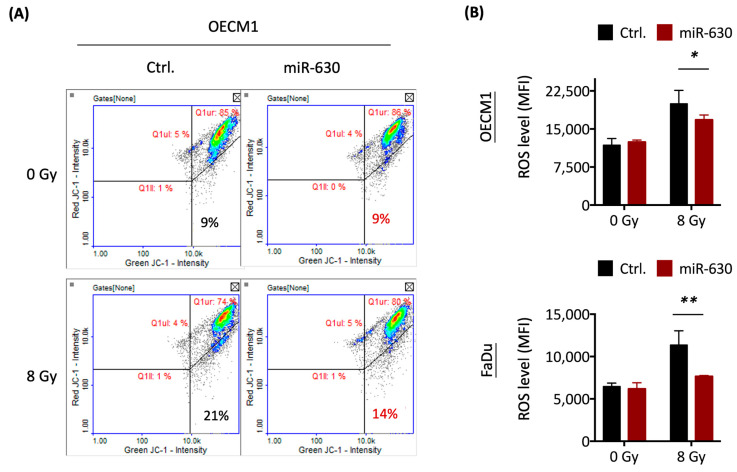
MiR-630 attenuates mitochondrial-mediated ROS level via Nrf2–GPX2 molecular axis. (**A**) MiR-630-attenuated mitochondrial damage in HNC cells following IR exposure. Representative image derived from image cytometry analysis of JC-1-stained cells isolated from OECM1 cells transfected with miR-630 mimic (miR-630) or control (Ctrl.) before 0 or 8 Gy IR treatment exposure for 24 h. (**B**) The mean fluorescentintensities (MFI) of ROS levels were determined by H_2_DCFDA staining in OECM1 and FaDu cells after 0 or 8 Gy IR exposure for 24 h. Data are presented as mean ± SD of three independent experiments (*, *p* < 0.05; **, *p* < 0.01; *t*-test). (**C**) MiR-630 enhanced Nrf2 transcriptional activity. Upper: A schematic representation of pGL3 promoter-8xARE construct, which was identified for measuring Nrf2 transcriptional activity. Lower: Luciferase reporter assay was used to compare the relative level of Nrf2 transcriptional activity in the HNC cells (OECM1 and FaDu) transfected with miR-630 mimic versus control transfectant. Data are presented as mean ± SD of three independent experiments. (*, *p* < 0.05; **, *p* < 0.01; *t*-test). (**D**) The protein levels were assessed using Western blot to detect Nrf2 in the miR-630 overexpressed or control cells. The β-actin was used as a loading control. (**E**) MiR-630 increased antioxidant enzyme GPX2. RT-qPCR analysis was used to compare expression levels of antioxidant enzymes GCLC, GCLM, MnSOD, and GPX2 in the HNC cells (OECM1 and FaDu) transfected with miR-630 mimic versus control transfectant. All expression values were normalized against an internal control, GAPDH, and then normalized to the control transfectant. Data are presented as mean ± SEM of three independent experiments (**, *p* < 0.01; *t*-test). (**F**) MiR-630 induced the expression of GPX2 protein. The HNC cells (OECM1 and FaDu) were transfected with miR-630 mimic or control. The protein level of GPX2 was performed using Western blot to detect the levels of the antioxidant enzyme at 24 h. The β-actin was used as a loading control. Data are presented as mean ± SD of three independent experiments (*, *p* < 0.05; **, *p* < 0.01; *t*-test).

**Figure 6 cells-12-02853-f006:**
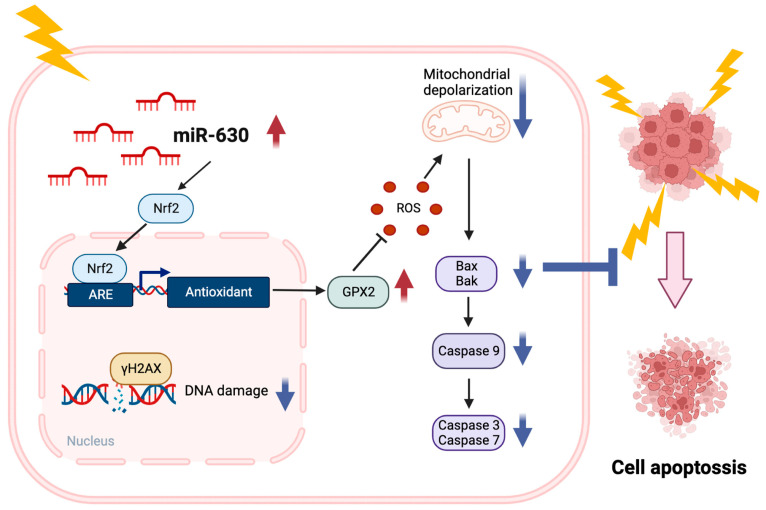
Schematic model of how miR-630 contributes radioresistance in HNC. A molecular model representing the miR-630 regulatory mechanism resulting in radioresistance in HNC cells is as follows: when miR-630 is overexpressed, it activates the Nrf2 molecule, subsequently causing an increase in the antioxidant enzyme GPX2. In turn, it reduces cellular levels of reactive oxygen species (ROS), which leads to less mitochondrial depolarization and diminished DNA damage from IR. These changes result in a decrease in the cellular intrinsic apoptotic response by inhibiting pro-apoptotic proteins (Bax and Bak) and reducing the activity of caspase 3/7/9 enzymes. Altogether, these molecular alterations confer radioresistance, which is associated with an unfavorable prognosis in HNC.

## Data Availability

The data presented in this study are available on request from the corresponding author.

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
