# Peer review of "MiR-630 Promotes Radioresistance by Induction of Anti-Apoptotic Effect via Nrf2–GPX2 Molecular Axis in Head–Neck Cancer"

_cells, 2023, doi:10.3390/cells12242853_

Round 1
Reviewer 1 Report
Comments and Suggestions for Authors
In Guo-Rung You et al‘s study, the authors report that MiR-630 promotes radioresistance by inducing anti-apoptotic mediated by the Nrf2-GPX2 axis in head & neck cancer.
My comments are listed as follows,
(1) The authors preferred to determine the miR-630 in the plasma from the HNC patients or healthy volunteers. How can the authors confirm that the miR-630 was released by the cancer cells? Why don’t directly detect the miR-630 in the primary tumors?
(2) It is also unclear what kind of HNCs were enrolled and how many samples from each HNC were selected.
(3) What is the target of miR-630? Which key molecule is regulated by miR-630? How this miR-630-regulated molecule affects the Nrf2-GPX2 axis.
(4) Regarding the IR dose, 2-6 Gy was used in Figure 2, 8 Gy was used in Figure 3 and Figure 5, and 6 Gy was used in Figure 4. How to decide the IR dose used for the experiments?
(5) Hava the authors tried to establish an in vivo model (like nude mice) to prove the hypothesis proposed here.
Author Response
To Reviewer and Editor:
We thank the reviewers’ comments for helping to make this manuscript a more meaningful document. Each comment has been well taken. We have made the point-by-point response to the reviewer's questions and amended it according to the suggestions. We have revised our manuscript with the "track changes" function in the text. Also, we have reread the article carefully and fixed a few grammatical errors. Your consideration for publication is very appreciated.
"Please see the attachment."

Reviewer 2 Report
Comments and Suggestions for Authors
The study titled "MiR-630 Promotes Radioresistance by Induction of Anti-Apoptotic Effect via Nrf2-GPX2 Molecular Axis in Head-Neck Cancer" offers a significant contribution to the understanding of head and neck cancer (HNC). The implications of this study are profound and offer a potential avenue for enhancing the effectiveness of radiotherapy in HNC.
By identifying miR-630 as a key player in radioresistance and elucidating its mechanism of action, the study sets the stage for further exploration and the development of targeted therapies that could positively impact patient outcomes.
From my point of view, I would recommend accepting this paper in its current form.
Comments on the Quality of English Language
Minor editing of English language required.
Author Response

(The authors gave the same response as above.)

Reviewer 3 Report
Comments and Suggestions for Authors
Excellent article. Certainly of interest to radiation oncologists and others. Although I am not a researcher in this field I am familiar with miRNAs and think the methodology is explained well, and the general conclusion that miR-630 is of interest with respect to radioresistance in HNC is sound.
Author Response

(The authors gave the same response as above.)

Reviewer 4 Report
Comments and Suggestions for Authors
Review of MiR-630 Promotes Radioresistance by Induction of Anti-Apoptotic Effect via Nrf2-GPX2 Molecular Axis in Head-Neck Cancer
I have completed my review of manuscript cells-2720299, entitled, “MiR-630 Promotes Radioresistance by Induction of Anti-Apoptotic Effect via Nrf2-GPX2 Molecular Axis in Head-Neck Cancer.”
This study investigates the role of microRNA-630 (miR-630) in radioresistance in head and neck cancer (HNC). miR-630, identified as an oncomiR, is overexpressed in HNC patients, correlating with a poorer prognosis. While miR-630 has minimal impact on cell growth, it significantly contributes to radioresistance in HNC cells by reducing cellular apoptosis, caspase enzyme activities, and irradiation-induced DNA damage, as indicated by decreased levels of the γ-H2AX histone protein. Mechanistically, miR-630 overexpression decreases cellular reactive oxygen species (ROS) levels, activating the Nrf2-GPX2 pathway, and leading to the upregulation of the antioxidant enzyme GPX2. The study establishes that miR-630 enhances radioresistance by inducing an anti-apoptotic effect through the Nrf2-GPX2 molecular axis in HNC. Targeting miR-630 may represent a promising strategy for radiosensitization in refractory HNC, providing insights for improved radiotherapy approaches.
The subject and results are interesting and useful. However, some modifications required in the present form of the manuscript.
Comments for authors
Comment 1: The introduction section of the manuscript provides a concise overview of the study's focus on microRNA-630 (miR-630) and its implications in radioresistance for head and neck cancer (HNC). However, for the benefit of new readers, expanding the introduction would be beneficial. Consider providing a more detailed background on the global prevalence of HNC, emphasizing the critical role of radiotherapy in treatment, and the challenges posed by radioresistance leading to local recurrence.
Comment 2: Given the observed correlation between miR-630 overexpression and poorer prognosis in HNC, elaborate on the potential downstream targets of miR-630 that contribute to cancer progression and treatment resistance.
Comment 3: What is the main mechanism for how utilization of miR-630 contributed to reducing the ROS levels? This needs to be explained in the manuscript.
Comment 4: I recommend discussing the role of ROS after irradiation in light of a recent study [https://doi.org/10.3389/fcell.2023.1067861] in the discussion section, lines 465 – 474.
Comment 5: The paper contains errors and typos. I encourage authors to reread carefully and fix any grammatical errors.
Comments on the Quality of English LanguageThe paper contains errors and typos. I encourage authors to reread carefully and fix any grammatical errors.
Author Response

(The authors gave the same response as above.)

Round 2
Reviewer 1 Report
Comments and Suggestions for Authors
The authors have thoroughly responded to my comments and revised the manuscript accordingly. I have no further concerns.
Author Response
To Reviewer 1:
We greatly appreciate your insightful comments, which have significantly improved the manuscript. We have carefully addressed each comment, incorporating the suggested changes using the 'track changes' function in the text. Your thoughtful consideration for publication is immensely appreciated.
Sincerely,
Guo-Rung You, Ann-Joy Cheng and Joseph T. Cheng
Chang Guan University and Linkou Chang Gung Memorial Hospital,
Taoyuan, Taiwan
Answer the specific questions.
Comment 1: The authors have thoroughly responded to my comments and revised the manuscript accordingly. I have no further concerns.
Response 1: Thank you for your feedback and for confirming that our revisions have addressed your concerns. We appreciate your contributions to refining our work.
Reviewer 2 Report
Comments and Suggestions for Authors
I do not have any more suggestions.
Author Response
To Reviewer 2:
We greatly appreciate your insightful comments, which have significantly improved the manuscript. We have carefully addressed each comment, incorporating the suggested changes using the 'track changes' function in the text. Your thoughtful consideration for publication is immensely appreciated.
Sincerely,
Guo-Rung You, Ann-Joy Cheng and Joseph T. Cheng
Chang Guan University and Linkou Chang Gung Memorial Hospital,
Taoyuan, Taiwan
Answer the specific questions.
Comment 1: I do not have any more suggestions.
Response 1: Thank you for your review and for confirming that there are no further suggestions. We appreciate your valuable input.
Reviewer 3 Report
Comments and Suggestions for Authors
I very much liked the original manuscript. The changes to it (other than minor grammatical corrections) add some useful information (eg PTEN) when it pertains to your study.
In other places (such as in the Intro) changes add nothing of value and I would delete. For example, details of when irradiation is used is irrelevant to your topic (and what you said was not entirely accurate).
Author Response
To Reviewer 3:
We greatly appreciate your insightful comments, which have significantly improved the manuscript. We have carefully addressed each comment, incorporating the suggested changes using the 'track changes' function in the text. Your thoughtful consideration for publication is immensely appreciated.
Sincerely,
Guo-Rung You, Ann-Joy Cheng and Joseph T. Cheng
Chang Guan University and Linkou Chang Gung Memorial Hospital,
Taoyuan, Taiwan
Answer the specific questions.
Comment 1: I very much liked the original manuscript. The changes to it (other than minor grammatical corrections) add some useful information (eg PTEN) when it pertains to your study. In other places (such as in the Intro) changes add nothing of value and I would delete. For example, details of when irradiation is used is irrelevant to your topic (and what you said was not entirely accurate).
Response 1: Thank you for your constructive feedback and for your positive remarks about the original manuscript. In response to your comments, we have revised the Introduction, reinstating some of the original text. We have also removed less relevant clinical details and incorporated additional molecular information pertinent to our study's topic (page 2 line 61-84). Your guidance has been invaluable in enhancing the quality and relevance of our manuscript.
Reviewer 4 Report
Comments and Suggestions for Authors
The authors have addressed all of my comments and concerns in the revised version. I recommend accepting the manuscript for publication in its present form.
Author Response
To Reviewer 4:
We greatly appreciate your insightful comments, which have significantly improved the manuscript. We have carefully addressed each comment, incorporating the suggested changes using the 'track changes' function in the text. Your thoughtful consideration for publication is immensely appreciated.
Sincerely,
Guo-Rung You, Ann-Joy Cheng and Joseph T. Cheng
Chang Guan University and Linkou Chang Gung Memorial Hospital,
Taoyuan, Taiwan
Answer the specific questions.
Comment 1: The authors have addressed all of my comments and concerns in the revised version. I recommend accepting the manuscript for publication in its present form.
Response 1: Thank you for your approval and recommendation for publication. We greatly appreciate your guidance and support.